# Towards a universal dataset and metrics for training and evaluating table extraction models

**Brandon Smock**
Microsoft
Redmond, WA
brsmock@microsoft.com

**Rohith Pesala**
Microsoft
Redmond, WA
ropesala@microsoft.com

**Robin Abraham**
Microsoft
Redmond, WA
robinab@microsoft.com

## Abstract

Recently, interest has grown in applying machine learning approaches to the problem of table structure inference and extraction from unstructured documents. However, progress in this area has been challenging not only to make but to measure, due to several issues that arise in both training and evaluating such systems from labeled data. This includes challenges as fundamental as the lack of a single definitive ground truth output for a given input sample and the lack of an ideal metric for measuring partial correctness for this task. To address these we propose a new dataset, PubMed Tables One Million (PubTables1M), and a new class of metric, *grid table similarity* (GriTS). PubTables1M is nearly twice as large as the current largest comparable dataset, can be used for models across multiple architectures and modalities, and addresses issues such as ambiguity and lack of consistency in the annotations. We apply DETR [1] to table extraction for the first time and show that object detection models trained on images and bounding boxes derived from this data produce excellent results out-of-the-box for all three tasks of detection, structure recognition, and functional analysis. In addition to releasing the data, we describe the dataset creation process in detail to enable others to build on our work and to ensure forward and backward compatibility of this data for combining it with other datasets created for these tasks. It is our hope that this data and the proposed metrics can further progress in this area by serving as a single source of data for training and evaluation of a wide variety of models for table extraction.

## 1 Introduction

Tables are a compact, structured representation for storing data and communicating it in documents and other manners of presentation, such as PDF or images. In its presented form, however, a table may not and often does not explicitly represent its logical structure. This is an important problem, as without this structure information, a significant amount of data in presentation tables is unable to be used in downstream applications.

The end-to-end problem of inferring a table's structure from its presentation and converting it into a structured form is called table extraction. This problem is very challenging for automated systems, as noted by many [2–5], and can be difficult even for human annotators [6], due to the wide variety of formats, styles, and structures found in presented tables. One of the main challenges is inferring the separations between cells in the absence of ruling lines between them, as shown in the table in Figure 1.

Submitted to the 35th Conference on Neural Information Processing Systems (NeurIPS 2021) Track on Datasets and Benchmarks. Do not distribute.

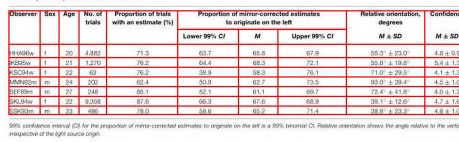

**TABLE 2** | Summary for the individual observers.

| Observer | Sex | Age | No. of trials | Proportion of trials with an estimate (%) | Proportion of mirror-corrected estimates to originate on the left | | | Relative orientation, degrees | Confidence |
|---|---|---|---|---|---|---|---|---|---|
| | | | | | Lower 99% CI | M | Upper 99% CI | M ± SD | M ± SD |
| HHA96w | f | 20 | 4,882 | 71.3 | 63.7 | 65.8 | 67.9 | 55.3° ± 23.0° | 4.8 ± 0.9 |
| IKB95w | f | 21 | 1,270 | 76.2 | 64.4 | 68.3 | 72.1 | 55.6° ± 19.8° | 5.4 ± 1.3 |
| KSC94w | f | 22 | 63 | 76.2 | 38.9 | 58.3 | 76.1 | 71.0° ± 29.5° | 4.1 ± 1.3 |
| MMN92m | m | 24 | 202 | 62.4 | 50.9 | 62.7 | 73.5 | 93.0° ± 39.4° | 4.5 ± 1.0 |
| SEF89m | m | 27 | 248 | 85.1 | 52.1 | 61.1 | 69.7 | 72.4° ± 41.8° | 4.0 ± 1.2 |
| SKL94w | f | 22 | 9,358 | 87.6 | 66.3 | 67.6 | 68.9 | 39.1° ± 12.6° | 4.7 ± 1.6 |
| SSK93m | m | 23 | 486 | 78.0 | 58.6 | 65.2 | 71.4 | 38.8° ± 23.3° | 4.8 ± 1.0 |

*99% confidence interval (CI) for the proportion of mirror-corrected estimates to originate on the left is a 99% binomial CI. Relative orientation shows the angle relative to the vertical irrespective of the light source origin.*

Figure 1: An example table without borders and ruling lines between cells.

(a) Ground truth as originally annotated          (b) Our preferred ground truth annotation

Figure 2: One challenge for creating ground truth for table structure recognition is that there are multiple ways to segment a table into cells that are compatible with its presentation.

Recently, there has been a shift in the research literature from traditional rule-based methods [7–9] for table extraction to data-driven methods based on deep learning (DL) [2, 10, 11]. The primary advantage of DL methods is that they can learn to be more robust to the wide variety of table presentation formats. However, these methods require a significant amount of data to train and thus far still rely significantly on additional rules, hand-engineered components, or special training procedures to achieve good performance.

Recent datasets for table structure recognition (TSR) [4, 3, 11], while large, have several limitations, including in some cases missing cell-level location information, compatibility with only specific model architectures, and lack of guarantees for data quality and consistency. A more fundamental issue, which we illustrate in Figure 5, is that for a given input table, there may not be only one way to annotate its structure [6]. Yet these datasets have been used for model training and evaluation as if each annotation is the only correct output, which leads to inconsistent feedback during training and noise during evaluation.

Another challenge for model evaluation in this area is the lack of an ideal metric. Several metrics have been proposed for evaluating the performance of TSR methods [12, 3, 13, 4]. While it is useful to have multiple metrics that evaluate TSR from different perspectives, these metrics lack a theoretical grounding, evaluate tables in ways that do not preserve their topological structure, and have different forms that lack an obvious connection between each other, making them difficult to interpret. Previous evaluations using these metrics have also not addressed the problem noted earlier, which is the possibility of multiple correct outputs for each input. This has made it difficult to benchmark current model progress, as it is not clear if when performance suffers it is due to deficiencies in the modeling or in the evaluation methodology.

To address these issues, we introduce a new dataset, PubMed Tables One Million (PubTables1M), and a new class of evaluation metric for table structure recognition, *grid table similarity* (GriTS).

- PubTables1M is the largest dataset of its kind. It contains nearly one million annotated tables from the PubMed Central Open Access (PMCOA) database, which is nearly twice as large as the current largest similar dataset, and nearly nine times as large as the most comparable dataset. It contains both PDF and image bounding box annotations for table detection, table structure recognition, and functional analysis, useful for training and evaluating any model whose data can be derived from PDF documents.

- As far as we know, PubTables1M is the first attempt to create a dataset with unambiguous ground truth for both training and evaluation, making it more suitable than previous datasets

for benchmarking progress in deep learning models. We introduce a canonicalization procedure whose goal is to ensure each table has a unique, unambiguous structure interpretation. We also process and filter the data to ensure it has consistent annotations for table content.

- Unlike previous metrics, grid table similarity (GriTS) evaluates a table in its natural matrix form. It also can evaluate multiple aspects of TSR within the same formulation, eliminating the need for different metrics that are difficult to compare.

- We apply the Detection Transformer (DETR) [1] for the first time to the tasks of table detection, structure recognition, and functional analysis, and demonstrate how with our data all three tasks can be addressed within an object detection framework out-of-the-box without the need for any custom components or training procedures.

- We plan to release all data and code for training and evaluation, which we hope will enable others to build off of and improve upon our work.

## 2   Background

Wang [14] distinguishes between a table in three forms, which we summarize here as:

1. Abstract table: a data structure that represents information in terms of a set of values, uniquely indexed by a multi-dimensional hierarchical system of keys.

2. Grid table: an abstract table with a two-dimensional arrangement of keys and values into cells occupying ordered rows and columns.

3. Presentation table: a concrete table; a visualization of a topological table with typography, spacing, and style.

A grid table is composed of cells, with each cell containing content. Each intersection of a row and a column forms a *grid cell*. A cell that spans multiple rows or multiple columns is called a *spanning cell*, and its content is considered to be repeated at each grid cell location that it spans.

Generally, table extraction (TE) is considered the problem of inferring a table's grid form from its presentation form. TE can be decomposed into three subproblems [15]: *table detection* (TD), which locates the table; *table structure recognition* (TSR), which recognizes the topological structure of a table in terms of rows, columns, and cells; and *functional analysis* (FA), which recognizes the keys and the values of the table. In this paper we address all three subproblems, but give particular attention to training and evaluating methods for TSR.

The output of a TSR system can be evaluated from three perspectives: *cell topology recognition*, which considers just the structure of the cells in a grid; *cell content recognition*, which considers both cell topology and the text content of each cell; and *cell location recognition*, which considers both cell topology and the absolute coordinates of each cell within a document. For evaluation, all three perspectives are useful. Cell content recognition is most aligned with the end goal of table extraction but for PDF and image input it can be dependent on the quality of OCR. Cell location recognition does not depend on OCR, but not every TSR method reports cell locations. Cell topology recognition is free of OCR and is applicable to all TSR methods, but is not anchored to the actual content of the cells either by text content or location. Thus, a high score on a cell topology metric would be necessary but not sufficient for performing well at table extraction.

## 3   Related Work

**Datasets**    Several large datasets have been introduced recently for table extraction [17, 18, 4, 3, 11]. We present an overview of recent datasets for TSR and compare the types of annotations they provide in Table 1. Among previous datasets for TSR, PubTabNet is the largest, with a total of 568k tables. The source data for PubTabNet are pairs of PDF and XML versions of the same scientific articles from the PMCOA database. PubTabNet is created through an automated matching process [18]

Table 1: Comparison of recent large datasets for table structure recognition.

| Name | Format | # Tables | Cell Topology | Cell Content | Cell Location | Canonical Ground Truth |
|------|--------|----------|---------------|--------------|---------------|------------------------|
| TableBank[4] | Image | 145k | ✓ | | | |
| SciTSR[16] | Image | 15k | ✓ | ✓ | | |
| PubTabNet[3] | Image | 568k | ✓ | ✓ | | |
| FinTabNet[11] | Image, PDF | 113k | ✓ | ✓ | ✓ | |
| PubTables1M (ours) | Image, PDF | 948k | ✓ | ✓ | ✓ | ✓ |

that for many tables in the XML can determine its corresponding bounding box in the PDF. While large enough to support training for deep learning models, it has some limitations, including that it lacks bounding box information for cells, only supports training and evaluation for specific model architectures, and only a small portion of the selected tables are considered complex, with any spanning cells. Without an explicit match between content at the individual cell level, there are also potentially unresolved issues with data quality. This is particularly a concern due to the use of a matching procedure and examples intended for table detection, which for that task can tolerate errors in cell-level annotations that then may go undetected for TSR.

**Metrics** Several evaluation metrics have been proposed for TSR. Göbel et al. [12] propose a content metric based on precision and recall for all pairs of adjacent cell content. Li et al. [4] propose a topology metric that evaluates HTML output with a custom tagset using the 4-gram BLEU score. Zhong et al. [3] propose a content metric that is a modified tree-edit distance on a custom HTML tagset and incorporates a text content score. Gao et al. [13] propose a location version of the metric proposed by Göbel et al. [12], which evaluates precision and recall for pairs of adjacent cells whose intersection-over-union (IoU) with a ground truth cell is above a threshold.

While it is useful to have multiple metrics that evaluate TSR from different perspectives, it is not obvious how these metrics relate to each other, making it unclear if a particular metric is best or how they should be used in combination. Each approximates a table as a set, a sequence, or a tree, none of which captures a table's two-dimensional structure. Both Zhong et al. [3] and Li et al. [4] also did not propose their metrics strictly for TSR, as they include aspects of functional analysis in their evaluations. These issues motivate us in Section 6 to propose new metrics with a clearer motivation that each retains a table's true topological structure and are natural to use in combination with one another.

# 4 PubTables1M Dataset

The source data for creating PubTables1M are pairs of PDF and XML versions of the same document from the PMCOA dataset. Roughly the same text appears in both, but the text in the PDF has spatial location $[x_{min}, y_{min}, x_{max}, y_{max}]$, while the text in the XML appears inside semantically labeled tags. We use the Needleman-Wunsch algorithm [19] to align the text from both sources, connecting each XML tag to its spatial location.

**Canonicalization** To remedy the issue of inconsistency and ambiguity in these annotations, we propose to convert each table annotation into a *canonical* form. This canonical form is similar to that defined by Seth et al. [20], who describe a set of permissible tilings of a table into cells. However, ours is motivated from the goal of ensuring each presentation table has a *unique interpretation*, which is a way of favoring one particular segmentation of table into rows, columns, and cells over other possibilities.

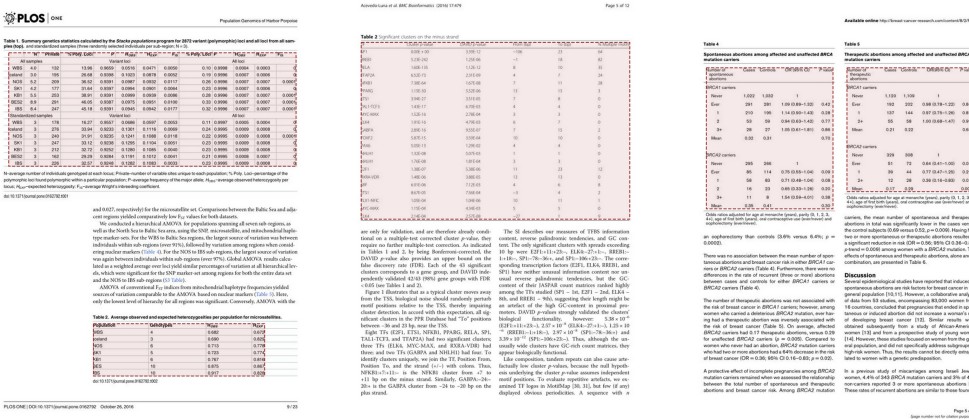

Figure 3: Examples of page images with table bounding box annotations in PubTables1M.

(a) Pre-canonicalization

(b) Post-canonicalization

Figure 4: The same table annotations before and after canonicalization.

To do this, our canonicalization procedure uses the idea that the row and column headers in a presentation table correspond in their abstract representation to trees. For an interpretation of the headers to be unambiguous, there should be a one-to-one correspondence between header cells and tree nodes. Canonicalization is a procedure to consolidate oversegmented header cells into a one-to-one correspondence with their abstract tree nodes. For the details of the procedure, please see the Appendix (code will be released).

**Header correction** The canonicalization procedure operates on cells in the row and column headers. The source XML annotations, however, do not label row headers, and we found that they sometimes contain incomplete annotations of the column headers, as well. Before canonicalization, we again use the assumption that the logical structure of the headers in their abstract representations is a tree to identify missing row header and incomplete column header annotations. Accurately labeling the full row header of a table for functional analysis is considered outside the scope of this paper. However, the high accuracy of our row header identification method is useful to correct oversegmented cells in the first column, leading to a significant net improvement in segmentation correctness for these cells.

There is one aspect of the row header, however, that is common enough and a special-enough case to include in both the canonicalization procedure and the annotations. This row header pattern has been referred to as a *projected multi-level row header* [21] or a *section header* [22]. An example of a table with a projected row header is given in Figure 4a. This is another common source of oversegmentation, as annotators differ on how to segment this row into cells. As each projected row header corresponds to one node in the tree representation of the header, we consolidate the entire row into a single spanning cell. For the tables in PMCOA, we consider this annotation of the spanning

cell as part of the row header accurate enough to include as part of the canonicalized ground truth. Figure 4b shows the table annotation after the full canonicalization procedure.

**Quality control**    Additional checks are needed to ensure the alignment locates content accurately and that the contents of the cells in their XML annotations match their PDF counterparts. For this, we discard any table annotations with rows that overlap each other, with columns that overlap each other, whose PDF cell contents do not match their XML annotations, or whose overall complexity is a significant outlier. For cell content, we check if the average edit distance between the PDF text content versus the XML text content in each corresponding cell is 0.05 or less. We choose not to force the text from each to be *exactly* equal, as the PDF text can differ even when everything is correct, due to things like word wrapping, which may add hyphens that would not appear in the XML. When the annotations do slightly differ, we choose to consider the PDF text to be the ground truth. For outlier removal, we measure complexity by the number of objects that are in the table, which is defined in Section 5, and cap the number of objects in a table at 100. In all, less than 0.1% of tables are discarded as outliers.

**Dataset splits and statistics**    Following the alignment, canonicalization, and quality control, from a large pool of documents we yield 947,642 annotated tables. Of these, 448,310 (47.3%) are simple tables and 499,332 (52.7%) are complex. Prior to canonicalization, only 379,735 (40.1%) of the tables in the set were considered complex by the original annotators. In total, canonicalization adjusts the annotations in some way for 328,421 tables (34.7%). 65.8% of the complex tables in the final set were adjusted from their original annotations. Finally, the method to add missing rows to the column header extends the header to more rows for 56,495 tables (6.0%).

We split the data randomly into train, validation, and test sets at the document level rather than the table level using an 80/10/10 split. For TSR, this results in 758,849 tables for training; 94,959 for validation; and 93,834 for testing. For each document, we note if all tables in the XML version of the document are present in the final set of annotations. While every table in the set can be used for training TSR models, only tables from documents with all of their tables annotated can be used for table detection. For TD, there are 460,589 fully-annotated pages containing tables for training; 57,591 for validation; and 57,125 for testing. The annotations are all on the source PDF documents themselves, which means they can be used for training any model whose data can be extracted from a PDF. However, one limitation of our implementation is we do not align tables that span multiple pages, so the data only contains tables that are fully contained within a single page.

# 5    Model

We model all three tasks of TD, TSR, and FA as object detection with images as input.

**TD model**    We use two object classes for TD: *table* and *table rotated*. The *table rotated* class corresponds to tables that are rotated counterclockwise 90 degrees, which is often the case for very wide tables. To create data for this model, we render the PDF pages to images with a maximum length of 1000 pixels and appropriately scale the bounding boxes for the objects to image coordinates.

**TSR and FA model**    We use a novel approach that models TSR and FA jointly using six object classes: *table*, *table column*, *table row*, *table column header*, *table projected row header*, and *table spanning cell*. The intersection of each pair of *table column* and *table row* objects can be considered to form a seventh implicit class, *table grid cell*. These objects model a table's hierarchical structure through physical overlap and model sequential ordering through their relative vertical and horizontal positioning. For TSR and FA, we first render the page containing the table as an image with a maximum length of 1000 pixels, scale and pad the table bounding box with an additional 30 pixels on all sides (or fewer on a side if there are less than 30 pixels available on that side), and crop the image to this bounding box. The padding enables more variation in training through cropping augmentations.

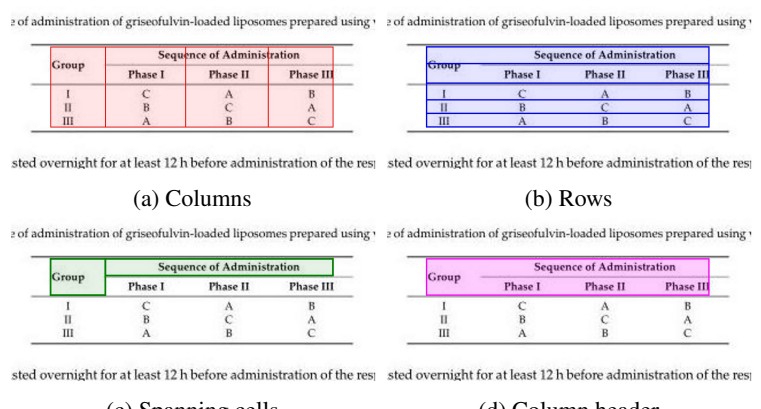

Figure 5: An example table with dilated bounding box annotations for different object classes.

**Dilated bounding boxes**  Besides adjusting the bounding boxes to their image coordinates, we make another adjustment just for the data for the TSR and FA model. For each pair of adjacent row bounding boxes and adjacent column bounding boxes, we expand their boundaries until they meet halfway, which fills the empty space in between them. After, there are no gaps or overlap between rows, and no gaps or overlap between columns. We call these *dilated* bounding boxes. We adjust the other objects so their boundaries match the adjustments made to the rows and columns they occupy.

**DETR**  To demonstrate the proposed dataset and the object detection modeling approach, we apply for the first time the Detection Transformer (DETR) [1] to all three table extraction tasks. We choose DETR over typical methods for object detection such as Faster R-CNN [23] due to DETR's superior ability to model global context for objects, as well as the fact that it does not perform an explicit early-stage non-maxima suppression step that would prevent it from outputting different classes with the same bounding box. We train one DETR model for TD and one model for TSR and FA. Each uses a ResNet-18 (R18) backbone, six layers in the encoder, and six layers in the decoder. For TD, we use 15 object queries, and for TSR and FA we use 125 object queries, each chosen to be slightly more than the maximum number of objects in each set's training samples. Besides this, we use the same default architecture settings for each.

**Additional components**  We use no custom components, losses, or procedures for training the model, other than standard data augmentations, such as random cropping and resizing. We only add a simple *conflict resolution* step used strictly at inference time, followed by a conversion step from the set of objects to a logical table. The conflict resolution step only involves removing objects or adjusting their bounding boxes to eliminate overlap between objects of the same class. For the sake of evaluation, we also align the bounding boxes to the text extracted from the document, though this action is taken after text extraction and has no effect on the outcome.

## 6   Proposed Metrics

To address the weaknesses of prior evaluation metrics, we propose a new family of related metrics we refer to as *grid table similarity* (GriTS). Unlike previous metrics, GriTS evaluates the topological representation of a table as a two-dimensional grid, or matrix.

**2D-LCS**  As a starting point for these metrics, we first consider the generalization of longest common substring to two dimensions, which is called two-dimensional longest common substructure (2D-LCS) [24]. Let $M[R, C]$ be a matrix with $R = [r_1, \ldots, r_m]$ representing its rows and $C = [c_1, \ldots, c_n]$ representing its columns. 2D-LCS operates on two matrices, $\mathbf{A}$ and $\mathbf{B}$, and determines the largest two-dimensional substructure, $\tilde{\mathbf{M}}$, the two have in common. In other words, $\tilde{\mathbf{M}} = \mathbf{A}[R'_A, C'_A] =$

246 $\mathbf{B}[R'_B, C'_B]$, where $R' \mid R$ is a subsequence of rows $R$, and $C' \mid C$ is a subsequence of columns $C$.

247 We can define a similarity score based on this as $S(\mathbf{A}, \mathbf{B}) = \frac{2|\tilde{\mathbf{M}}|}{|\mathbf{A}|+|\mathbf{B}|}$, where $|\mathbf{M}_{m \times n}| = m \cdot n$.

248 **2D-MSS** An extension to this is to relax the exact match constraint, and instead determine the two
249 most *similar* two-dimensional substructures, $\tilde{\mathbf{A}}$ and $\tilde{\mathbf{B}}$. We define this by replacing equality between
250 entries $\mathbf{A}_{i,j}$ and $\mathbf{B}_{i,j}$ with some choice of similarity function between them $f(\mathbf{A}_{i,j}, \mathbf{B}_{i,j})$, which
251 maps to the range $[0, 1]$. We call this two-dimensional most similar substructures (2D-MSS).

252 **Grid table similarity (GriTS)** GriTS is 2D-MSS with a particular choice of similarity function
253 and a particular matrix of entries to compare. Given a similarity function $f()$ and choice of matrices
254 $\mathbf{A}$ and $\mathbf{B}$ we define $GriTS_f$ as:

$$GriTS_f(\mathbf{A}, \mathbf{B}) = \max_{R'_A, C'_A, R'_B, C'_B} \frac{2 \cdot \sum_i \sum_j f(\mathbf{A}[R'_A, C'_A]_{i,j}, \mathbf{B}[R'_B, C'_B]_{i,j}))}{|\mathbf{A}| + |\mathbf{B}|}, \tag{1}$$

$$= \frac{2 \cdot \sum_i \sum_j f(\tilde{\mathbf{A}}_{i,j}, \tilde{\mathbf{B}}_{i,j}))}{|\mathbf{A}| + |\mathbf{B}|}. \tag{2}$$

255 One of the main advantages of GriTS is we can use the same formulation for all aspects of TSR.
256 We define one version for cell location recognition ($GriTS_{\text{Loc}}$), one for cell content recognition
257 ($GriTS_{\text{Cont}}$), and one for cell topology recognition ($GriTS_{\text{Top}}$). For cell location recognition, $\mathbf{A}$ and
258 $\mathbf{B}$ are such that $\mathbf{A}_{i,j}$ contains the bounding box of the cell located at row $i$ and column $j$. The function
259 we use for comparing the similarity of two bounding boxes is the standard intersection-over-union
260 (IoU). For cell content recognition, $\mathbf{A}$ and $\mathbf{B}$ are such that $\mathbf{A}_{i,j}$ contains the text content of the cell
261 located at row $i$ and column $j$. The function we use for comparing the similarity of two strings of
262 text content is normalized longest common substring (LCS).

263 For cell topology recognition, we use the same similarity function as cell location recognition, IoU,
264 but on bounding boxes with size and relative position given in the grid coordinate system. Let $\alpha_{i,j}$
265 be the rowspan of the cell at position $(i, j)$, let $\beta_{i,j}$ be the colspan of the cell at position $(i, j)$, let
266 $\rho_{i,j}$ be the minimum row occupied by the cell at position $(i, j)$, and let $\theta_{i,j}$ be the minimum column
267 occupied by the cell at position $(i, j)$. Then for cell topology recognition, $\mathbf{A}$ and $\mathbf{B}$ are such that $\mathbf{A}_{i,j}$
268 contains the bounding box $[\rho_{i,j} - j, \theta_{i,j} - i, \rho_{i,j} - j + \beta_{i,j}, \theta_{i,j} - i + \alpha_{i,j}]$. Note that for any cell
269 with rowspan of 1 and colspan of 1, this box is $[0, 0, 1, 1]$.

270 **Factored 2D-MSS** Computing the 2D-LCS of two matrices is NP-hard [24]. This suggests that all
271 metrics for TSR may end up being an approximation to what could be considered the ideal metric.
272 We propose a heuristic approach to determine the most similar 2D substructures by factoring the
273 problem and determining the optimal 1D subsequences of rows and of columns from each matrix
274 independently. This procedure uses dynamic programming (DP) in a nested manner, which is run
275 twice: once to determine the most similar rows and once to determine the most similar columns
276 between the two matrices. The nested DP procedure is $O(|\mathbf{A}| \cdot |\mathbf{B}|)$.

277 Because the outcome of the procedure is a selection of rows and columns for each matrix, it still
278 yields a valid 2D substructure of each; these just may not be the most similar substructures possible.
279 It follows that the similarity computed using this procedure is a lower bound on the true similarity
280 between $\mathbf{A}$ and $\mathbf{B}$.

## 7 Experiments

282 **Metrics** To validate the behavior of the proposed metrics, we perform experiments where we
283 evaluate each metric on the actual ground truth versus versions of the ground truth that are corrupted
284 in straightforward ways. To produce a corrupted version of the ground truth, we select and keep rows
285 and columns from the actual ground truth with probability $x$, where $x$ can vary from $[0, 1]$, while
286 keeping the rows and columns in their original order.

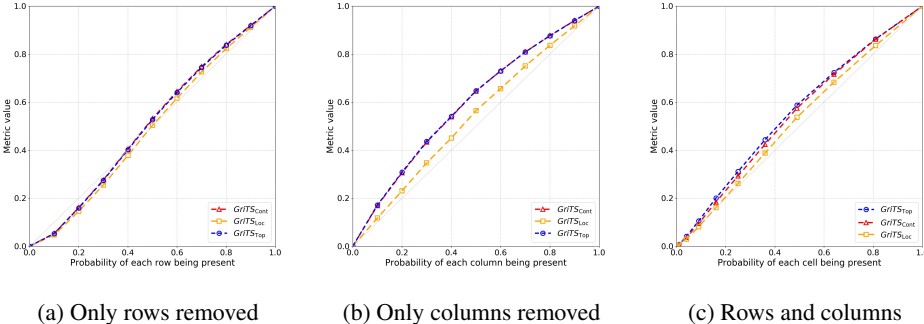

| (a) Only rows removed | (b) Only columns removed | (c) Rows and columns |

Figure 6: Comparison of GriTS for the ground truth versus corrupted ground truth where we keep each row, each column, or both (in their original order) with probability $x$.

Table 2: Test performance of both models on PubTables1M using object detection metrics.

| Model | Task | AP50 | AP75 | AP | AR |
|---|---|---|---|---|---|
| DETR-R18 | TD | 0.995 | 0.988 | 0.966 | 0.981 |
| DETR-R18 | TSR + FA | 0.971 | 0.948 | 0.912 | 0.942 |

We report three such experiments, one where we keep all columns but select rows to keep with probability $x$, one where we keep all rows but select columns to keep with probability $x$, and one where we select both rows and columns with probability $x$, which keeps each cell with probability $x^2$. In each experiment, we vary $x$ in increments of $0.1$. We report the results of these experiments in Figure 6. Since the rows and columns remain in their original order, $x$ can be interpreted as the expected value of the fraction of true rows and columns in the ground truth that are in their true order and $x^2$ as the expected value of the fraction of cells in a valid substructure of the true matrix of cells. For each experiment, this simulates evaluating the performance of a model that exhibits these expected values.

As can be seen in Figure 6, all of the metrics are closely related to the fraction of rows, columns, and cells reported by a model that appear *in the same order* as they appear in the ground truth in both directions of the table, which is their desired behavior. Taken together, these results validate that all of the metrics can distinguish between good and bad models, carry a straightforward interpretation when evaluating model performance, and closely relate to each other despite their different forms.

**Model Evaluation**   In the next set of experiments, we train each DETR-R18 model on the object detection data derived from PubTables1M. All of the experiments are performed using a single NVidia Tesla V100 GPU. We train each model for 20 epochs and use all default hyperparameters except for those we note here. For both models, we use a learning rate drop of 1 and gamma of 0.9. For the TSR and FA model, we also use an initial learning rate of 0.00005 and a no-object class weight of 0.4. We limited hyperparameter tuning to one short experiment to determine the initial learning rate. We ran training experiments with three different initial learning rates of 0.0002, 0.0001, 0.00005 and chose to use the learning rate for each model that had the best performance on the validation set after one epoch of training.

We report evaluation of the trained models on the full test set using both standard object detection metrics and the proposed GriTS metrics. The average precision (AP), AP50, AP75, and average recall (AR) of the two models is displayed in Table 2. In Table 3, we report the performance of the DETR-R18 TSR and FA model according to our proposed metrics. We report a breakdown of the results by type between simple tables, which have no spanning cells, and complex tables, which do. We use a confidence threshold of 0.5 for all classes. For evaluating our TSR model according to cell location recognition, we report the cell locations after the conflict resolution stage that, in addition

Table 3: Test performance of the TSR + FA model on PubTables1M on the proposed GriTS metrics.

| Data split | # Samples | $GriTS$ | | | |
|---|---|---|---|---|---|
| | | Top | Cont | Loc | RawLoc |
| Simple | 44,355 | 0.995 | 0.995 | 0.992 | 0.947 |
| Complex | 49,479 | 0.975 | 0.983 | 0.966 | 0.909 |
| All | 93,834 | 0.985 | 0.989 | 0.978 | 0.927 |

to removing overlap between objects of the same class, also adjusts the row and column bounding boxes to tightly surround the bounding boxes for the words they contain.

To assess how well the DETR-R18 TSR model performs with no post-processing, we define a fourth metric, $GriTS_{\text{RawLoc}}$. $GriTS_{\text{RawLoc}}$ uses the same similarity function as $GriTS_{\text{Loc}}$ but the matrix of predicted cell bounding boxes are the raw output of the model, which we compare to the true dilated bounding boxes. The difference between $GriTS_{\text{Loc}}$ and $GriTS_{\text{RawLoc}}$ mostly measures the impact of the conflict resolution stage on performance.

## 8  Conclusion

In this paper we introduced a new dataset, PubMed Tables One Million (PubTables1M), the largest of its kind, and *grid table similarity* (GriTS), a new class of evaluation metric for table structure recognition that has a much better theoretical grounding than previously proposed metrics. Pub-Tables1M is the first attempt to create a large-scale dataset for table structure recognition with consistent, unambiguous ground truth. Unlike previous metrics proposed for TSR, GriTS evaluates table structure recognition in multiple ways within the same formulation, and can do so in a table's natural matrix form. We trained DETR for the first time for the tasks of table detection, table structure recognition, and functional analysis, demonstrating excellent performance out-of-the-box using our data with minimal customization for these tasks. We believe PubTables1M and GriTS can further progress in this area by enabling for the first time the chance to train and compare models across different modalities and output formats with the same dataset and evaluation framework. While we do not believe this work raises any potential issues regarding negative impacts to society, we have documented the computation used in our experiments and noted any exclusions in our dataset that potentially could lead to impacts if incorporated into real-world systems. We welcome a discussion on any additional potential impacts raised by others.

## 9  Future Work

We hope the dataset and metrics proposed in this paper will aid progress by making it much easier to compare different methods for table extraction in the future. While the tables derived from scientific articles are diverse, we think it could be very useful to apply the canonicalization and quality control procedures proposed in this work to additional datasets for table extraction to increase the variety of training data and evaluation generalization across document types. Finally, we believe releasing a large collection of high-quality data samples for table extraction is helpful not just for that isolated task but also provides a large starting pool of data for combining with annotations for additional tasks made on the same source data. Consolidating document parsing tasks from across multiple sets of data and labels represents an interesting direction for work in this area and is something we plan to pursue in the future.

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
