# OpenReview forum: "Towards a universal dataset and metrics for training and evaluating table extraction models"
_NeurIPS.cc/2021/Track/Datasets_and_Benchmarks/Round1 — Submitted to NeurIPS 2021 Datasets and Benchmarks Track (Round 1)_

### Official Review · Reviewer_NfFe · 2021-07-04
**A solid step forward towards a useful benchmark and dataset**

**Rating:** 7
**Confidence:** 3
**Correctness:** The dataset and benchmark both seem s…
**Clarity:** The writing is clear and well-organiz…

**Strengths:**

The authors provide a large-scale dataset towards an important problem which seems relevant to the broader information extraction research community. The corresponding metrics seem reasonable measures of progress towards this task, and the baseline model established demonstrates high performance.

**Weaknesses:**

I would have liked to see more of a discussion on the practical implications of models that do better or worse on the metrics the authors propose. Are there certain types of data that are frequently lost? Are there certain types of tables that will be better or worse served, depending upon how a model does on the various metrics?

Given the baseline's apparent strong performance, I'd also like a better understanding of what the authors see as the most important remaining directions in this space.

**Additional Feedback:**

N/A

**Documentation:**

The supplemental material documents the dataset and its collection process well.

**Ethics:**

I'd be interested to see some sort of evaluation as to whether the type of information presented in tables is correlated with better or worse performance (e.g. perhaps tables discussing demographic information tend to be structured in a way that the author's baseline struggles with).

**Relation To Prior Work:**

I'm not an expert in prior work in this area, but existing work seems well-cited and clear differentiations are made.

**Summary And Contributions:**

The authors present a large-scale dataset and metrics for determining the structure of tables in unstructured documents. The authors tackle an important problem, and the dataset the propose, the metrics they propose, and the baseline model that they propose all seem like significant steps forward.

---

> ### Author Response · Authors · 2021-07-14
> **Response and thanks to reviewer #3**
>
> First, we would like to express our deep thanks to the reviewer for investing the time to thoroughly evaluate our work and make suggestions for improvements to the paper.
>
> > I would have liked to see more of a discussion on the practical implications of models that do better or worse on the metrics the authors propose. Are there certain types of data that are frequently lost? Are there certain types of tables that will be better or worse served, depending upon how a model does on the various metrics?
>
> Thanks for the nice suggestion. In response we have added an analysis of the behavior of the metrics that we believe is intuitive and makes it clearer how the metrics behave and relate to each other. Important to note is that one of our primary motivations for having the different metrics is to make it easier to directly compare models that output table structure in different ways. So in this regard it would be ideal if each of the metrics would perform similarly in response to different kinds of tables. But we agree that if there were types of tables where the metrics diverged then this would provide additional useful information about an individual model’s behavior. We will plan experiments that explore how the metrics compare on different table types, such as simple versus complex tables or tables with variable row widths versus those with consistent row widths, and add them to the appendix in the camera-ready version of the paper.
>
> > Given the baseline's apparent strong performance, I'd also like a better understanding of what the authors see as the most important remaining directions in this space.
>
> Thanks for the suggestion. We agree this is important and in response we have used the additional page allowed to us to add a Future Work section that outlines some of our initial thoughts.
>
> > I'd be interested to see some sort of evaluation as to whether the type of information presented in tables is correlated with better or worse performance (e.g. perhaps tables discussing demographic information tend to be structured in a way that the author's baseline struggles with).
>
> While previous work on large-scale datasets only breaks down the tables by simple vs. complex, as we have done, we agree that it is an interesting research question to understand other ways in which the tables can be categorized and performance looked at over the categories. We think the particular suggestion the reviewer made is likely outside the scope of the paper and is a question for future work, unless we could obtain presently very reliable labels for such types of tables. But for the camera-ready version of the paper we will consider all ways we can reliably break down the performance of the model by table category, including perhaps small tables (with a few rows) versus large tables, or tables with equal-width rows versus tables with highly-variable-width rows, and add the performance breakdown to the experimental results, which we think will be a valuable addition.

---

> > ### Comment · Reviewer_NfFe · 2021-07-20
> > **Thanks**
> >
> > Thanks to the authors for their response! I remain positive about this paper.

---

### Official Review · Reviewer_vyFL · 2021-07-04
**A dataset and metrics for training and evaluating table extraction models**

**Rating:** 4
**Confidence:** 2

**Strengths:**

The paper seems to represent an improvement in the state-of-the-art for training ML methods to infer and extract structure and content from tables in PDF files.

**Weaknesses:**

For people unfamiliar with prior literature, the paper does not do a great job of motivating why the problem of table extraction is challenging.

The contributions here seem very incremental. It appears that the dataset was generated using almost exactly the same procedure as before but over large publication data corpus. The big change appears to be in terms of creating ambiguous ground truth (via a canonicalization procedure), but here it is unclear what the key new insight behind the procedure is or what the drawbacks of such canonicalization might be.

Similarly, for someone unfamiliar with prior literature on the topic, it is unclear how the proposed metrics have addressed the drawbacks of the existing metrics.

**Additional Feedback:**

I would encourage the authors to more clearly demonstrate the benefits of the dataset, the canonicalization procedure, and the proposed metrics.

**Clarity:**

At one level, the paper is easy to read and follow. But, the paper is written for audience already familiar with the literature on table extraction models. For example, as someone unfamiliar with the literature, the review found it is very hard to understand the drawbacks with existing metrics from the description here.

**Correctness:**

It is unclear how the increase (doubling) in size of the dataset benefitted the models trained. Similarly, it is unclear whether and how the new metrics helped address the drawbacks with existing metrics.

**Documentation:**

Yes

**Relation To Prior Work:**

The authors do a good job of discussing related work and how the work improves on previous contributions. But, the contributions strike the reviewer as too incremental.

**Summary And Contributions:**

The paper focuses on creating a dataset and metrics for better training and evaluation of methods for extraction table structures.

It claims that the dataset is larger (almost double) that of previously generated datasets and has consistent annotations. Though the methodology for creating the dataset seems identical to approaches used before.

---

> ### Author Response · Authors · 2021-07-14
> **Response and thanks to reviewer #2**
>
> First, we would like to thank the reviewer for investing the time to familiarize themselves with a new research area and provide valuable feedback to which we give careful consideration and address with several improvements.
>
> > For people unfamiliar with prior literature, the paper does not do a great job of motivating why the problem of table extraction is challenging.
>
> We appreciate the feedback on this and to address this we have added more citations in the introduction to papers that both highlight and explore the difficulty of the problem.
>
> > It is unclear how the increase (doubling) in size of the dataset benefitted the models trained.
>
> To address this, we will add results from an experiment in the camera-ready version of the paper training DETR on 1/10th of the data and evaluating it on the full test set, which will help to answer how important the size of the dataset is. While this is important, we believe in this case that dataset size has more potential benefits than just those that arise from training for the tasks in this paper. To further highlight the significance of dataset size, we mention one of these additional benefits in the Future Work section, which we have added in the most recent revision.
>
> > The contributions here seem very incremental. It appears that the dataset was generated using almost exactly the same procedure as before but over large publication data corpus.
>
> We appreciate the feedback here. However, we think the paper already clearly differentiates our contributions from those prior and explains why these are significant.
>
> It is true that prior work has also attempted to create a large-scale dataset for table extraction using alignment between PDF and XML annotations. But not only is our alignment process more effective, we improve upon the entire dataset creation process in several novel and significant ways.
>
> To understand why even our alignment process is a significant improvement over previous attempts at alignment, note that the authors of the PubTabNet dataset state in their paper that only a small portion of the tables in their dataset were complex (having any spanning cells). On the other hand, 40.1% of the tables that are yielded by our alignment process and   make it through subsequent filtering stages were complex even without canonicalization, making our dataset much richer and more useful even before any of the more novel aspects of our paper are considered. We will take the opportunity to call more attention to this in the camera-ready version of the paper, which we think will help to address the reviewer’s feedback here.
>
> In terms of what the paper achieves versus prior work, note that no prior work attempted to make their ground truth unambiguous or to use a metric to precisely control the quality of the annotations. This makes it unclear for those wanting to put the data from prior work to use how much noise  is retained within the data. As we state in the paper, we use the edit distance between the aligned text in every individual cell between the XML and the PDF versions of the table to ensure the annotations are consistent/correct. Any prior dataset that does not report the bounding box of every cell cannot even attempt to do this quality control step. In other words, unlike previous work, our dataset comes with measurable quality guarantees.
>
> Note that without a measurable quality control step, large-scale datasets in prior work can specifically retain mistakes from the original annotators and retain mistakes from the imperfect alignment process. Without canonicalization, large-scale datasets can have significantly inconsistent ground truth for complex tables. We quantify this problem by pointing out that our canonicalization and correction procedure changed the annotations from their original structure labels for 65.8% of the complex tables in the dataset, which is hugely significant. We illustrate such differences in Figures 2 and 4. Our contribution, therefore, addresses a significant source of noise in the original annotations. It is even more vital that we do this in our dataset versus previous datasets since we have a higher percentage of complex tables. We will take the opportunity to emphasize these points in the camera-ready version of the paper.
>
> While the above already well differentiates our work from prior work, we believe we can address any remaining concern the reviewer might have by including additional experiments that quantify further some of the individual contributions. For the camera-ready version of the paper, we will add an experiment that trains the proposed DETR model on a smaller version of the dataset, which will assess the direct impact of having more data. Further we will add an experiment that trains DETR on the dataset without any canonicalization or correction procedures and compare the results with the model reported in the current paper. This will further quantify the importance and the impact of these steps.

---

> ### Author Response · Authors · 2021-07-14
> **Additional response to reviewer #2**
>
> > The big change appears to be in terms of creating ambiguous ground truth (via a canonicalization procedure), but here it is unclear what the key new insight behind the procedure is or what the drawbacks of such canonicalization might be.
>
> We believe we address the first part of this comment, the novelty and impact of the canonicalization and quality control procedures, in the previous response above.
>
> We think the reviewer makes a nice point here about considering the drawbacks of canonicalization. In the paper we motivate canonicalization as theoretically the ideal thing to do, as each table should have a unique abstract representation. From this point of view, there are no drawbacks. But we think we need to add that our canonicalization procedure itself may be imperfect, or in some cases there may not be enough information in the annotation to recover a unique abstract representation. This is not necessarily a drawback but important to note, and we will add this to the camera-ready version of the paper.
>
> > …it is unclear whether and how the new metrics helped address the drawbacks with existing metrics.
>
> > as someone unfamiliar with the literature, the review found it is very hard to understand the drawbacks with existing metrics from the description here.
>
> > I would encourage the authors to more clearly demonstrate the benefits of the dataset, the canonicalization procedure, and the proposed metrics.
>
> We thank the reviewer for the feedback and suggestions. We agree that more can be done here and believe the above explanation and the proposed improvements to the experiments should address the reviewer’s feedback about more clearly demonstrating the benefits of the dataset and canonicalization procedure.
>
> In terms of the benefits of the proposed metrics, the first problem we point out with previous metrics in the paper is that they come with no theoretical grounding or justification. On the other hand, we start from the idea that the ideal metric would be a similarity between matrices, and then we take steps to derive such a metric that is also not intractable to compute. As noted in the paper, previous metrics represent grid tables as sets, sequences, and trees and then compute similarity between these objects, which clearly will have different properties than similarity between matrices. Since our metric is based on matrix similarity, it preserves the topology of the cells when computing similarity. These theoretical considerations alone mark a clear improvement in our proposed metrics over prior ones.
>
> To help demonstrate the benefits of the proposed metrics empirically, in the current version we add an experiment that shows that the metrics have a nice interpretation and exhibit behavior consistent with expectations for a similarity metric between tables. In the camera-ready version of the paper, we will either provide examples that better illustrate the differences between the proposed metric and previous metrics or add an experiment that directly compares the behavior empirically, which we believe addresses the feedback from the reviewer.

---

### Official Review · Reviewer_ar1n · 2021-07-05
**Review (Updated 19th July 2021)**

**Rating:** 7
**Confidence:** 3
**Correctness:** Seems good

**Strengths:**

- providing a large dataset for a relevant task
- the overall way the dataset is created is documented
- the dataset will be accessible (via Microsoft Research Open Data repository) and accountability is provided
- proposing a new performance metric and a canonical form of the label

**Weaknesses:**

- the actual steps of the dataset generation process are unclear
- the experiments do not help to support the claims made in the paper

**Additional Feedback:**

### Dateset generation process
Given the focus of this track, a clearer (i.e., reproducible) description of the data generation process would be desirable. You provide code for the experiments. Hence, you could easily give the code for the dataset generation or at least for the processing.

### Experiments
The experiments use a new technique for training on a new dataset using a new performance metric. In the current form, it is impossible to put the results into perspective. You say you get good results, but given that this is a new metric, it is unclear what good results mean. Maybe the new metric is just close to 1 for everything. The experiments would be more meaningful if they would explore each new component individually. To demonstrate the metric's usefulness, you could show that it can differentiate between a good and a bad model. Additionally, you could demonstrate alignment with performance in a downstream task. To demonstrate the usefulness of DETR, compare it to existing techniques. Finally, to demonstrate the dataset's usefulness, show how training on existing datasets (or a smaller version of this dataset) leads to worse performance.

### Potential bias
The dataset is trying to solve the table extraction task, but the creation process might introduce bias. First, all tables come from a specific domain (medical publications). Second, the original database required the submission of XML versions of the tables. This requirement might lead the authors of the papers to restrict themselves in what type of table they create. These factors might limit the generalization of models trained on this dataset to other domains. While these biases don't seem to raise ethical concerns, I recommend adding these potential problems in the datasheet and maybe clarify them in the paper.

**Clarity:**

Overall, the paper is well structured and provides enough background for a reader unfamiliar with the task of table extraction. The process of creating the dataset is also explained. However, the individual steps are a little unclear. Especially the canonicalization procedure used to create unique labels. The paper doesn't describe how the procedure works and does not provide any code for this step. Given that the procedure is mentioned as one of the major contributions by the authors, it would be good to explain it. The header correction step is also a little confusing. It mentions a "same assumption" (l. 151) and it's unclear what assumption that is. This can easily be clarified, by either providing a step-by-step description of the process or including the code used for these steps.

**Documentation:**

Overall yes. Only the processing step from raw data to the dataset should be clarified.

**Ethics:**

No. The dataset is a transformation of publicly available data from scientific articles.

**Relation To Prior Work:**

The paper mentions both prior datasets as well as performance metrics and describes their differences. It doesn't empirically compare to existing work.

**Summary And Contributions:**

**Update**
In the original review, I stated that the main weaknesses of the paper seem to be easily fixable, given the response from the authors I think my main concerns have been addressed. I've updated my score accordingly.

The paper introduces a new large dataset for the task of table extraction. The dataset is extracted from articles in the PubMed Central Open Access database. Further, the paper proposes a new way to create unambiguous ground truth for table extraction and a new performance metric for the task. The paper provides background for the underlying task and an overview of the creation of the dataset. Finally, the authors conduct some experiments using the new dataset, performance metric, and a new model.

---

> ### Author Response · Authors · 2021-07-14
> **Response and thanks to reviewer #1**
>
> First, we would like to thank the reviewer for investing the time to provide very useful feedback and thoughtful suggestions for how to improve the paper. We think the final paper will be much stronger after incorporating the reviewer’s very clear suggestions for improvement.
>
> > the actual steps of the dataset generation process are unclear
>
> > Only the processing step from raw data to the dataset should be clarified.
>
> > …the individual steps are a little unclear. Especially the canonicalization procedure used to create unique labels. The paper doesn't describe how the procedure works and does not provide any code for this step . Given that the procedure is mentioned as one of the major contributions by the authors, it would be good to explain it.   The header correction step is also a little confusing. It mentions a "same assumption" (l. 151) and it's unclear what assumption that is. This can easily be clarified, by either providing a step-by-step description of the process or including the code used for these steps.
>
> > Hence, you could easily give the code for the dataset generation or at least for the processing.
>
> We agree with the reviewer that including the full steps of the procedure is important because the canonicalization process is one of the major contributions of the paper, and full details were omitted originally only due to page constraints. In response, we have included an appendix in the current supplementary materials where we describe the canonicalization and correction procedure step-by-step. In addition, we plan to release the code for the entire dataset creation process along with the camera-ready version of the paper, which we agree ensures reproducibility and strengthens the contribution made by the paper.
>
> We also clarify the quote from Line 151 in the current revision.
>
> > the experiments do not help to support the claims made in the paper
>
> > given that this is a new metric, it is unclear what good results mean.  Maybe the new metric is just close to 1 for everything. The experiments would be more meaningful if they would explore each new component individually. To demonstrate the metric's usefulness, you could show that it can differentiate between a good and a bad model.
>
> To address the reviewer’s feedback and suggestions, we have used the additional allowed page to include additional experiments in the current revision that show the metrics are well-behaved and well-distinguish good models from bad models. These experiments provide an intuitive interpretation for the actual values of the metrics in terms of the expected recall of both rows and columns in their correct order. Together, these results provide direct evidence that models must exhibit desirable behavior to score high on the proposed metrics, which supports the claim made in the paper that DETR’s performance is good.
>
> > To demonstrate the usefulness of DETR, compare it to existing techniques.
>
> As the focus of the paper is on the dataset and metrics, we believe a full comparison of DETR with existing techniques would be out of the scope of the paper. However, we thank the reviewer for this suggestion and agree that it would be helpful to have a comparison with another technique. To incorporate the suggestion and to show that DETR’s strong performance does not indicate that the proposed dataset is trivial to learn, in the camera-ready version of the paper we will add an experiment training Faster R-CNN on the proposed data and comparing the performance with DETR. Faster R-CNN has been used previously for table detection and for some aspects of table structure recognition and can be trained on the same data as DETR, making for a fair and useful comparison.
>
> > …to demonstrate the dataset's usefulness, show how training on existing datasets (or a smaller version of this dataset) leads to worse performance.
>
> We thank the reviewer for this suggestion and in the camera-ready version of the paper we will add two more experiments to address this and measure the usefulness and impact of different aspects of the proposed dataset: 1. Training DETR on 1/10th of the data and evaluating it on the full test set, which will help to answer how important the size of the dataset is; 2. Training DETR on the unprocessed (non-corrected, non-canonicalized) annotations, which will help to show how much the canonicalization and correction process contributes to reducing noise in the training and evaluation.

---

> ### Author Response · Authors · 2021-07-14
> **Additional response to reviewer #1**
>
> > The dataset is trying to solve the table extraction task, but the creation process might introduce bias. First, all tables come from a specific domain (medical publications). Second, the original database required the submission of XML versions of the tables. This requirement might lead the authors of the papers to restrict themselves in what type of table they create. These factors might limit the generalization of models trained on this dataset to other domains. While these biases don't seem to raise ethical concerns, I recommend adding these potential problems in the datasheet and maybe clarify them in the paper.
>
> We agree with the reviewer and agree that it is important to note that the tables in this dataset, while diverse, may not reflect the distribution of tables in other types of documents. We have updated the datasheet as the reviewer suggested and added a note in the Future Work section to call attention to the additional research needed here.

---

### Author Response · Authors · 2021-07-14
**Response and thanks to all reviewers, and improvements made in the current revision**

First, we would like to express our deep gratitude to the reviewers for taking the time to review our work and provide valuable feedback. We notice the unique perspective each reviewer brought to the process and we are grateful for the opportunity to strengthen our paper in response. We also recognize the extra time that the reviewers invested to get more familiar with the relevant literature and want to express our great appreciation.

We address specific feedback from each reviewer in their own individual responses, but here we address a few of the major points raised and discuss the improvements we have made in response in the most recent revision.

Before we address the feedback and discuss improvements, we would like to add a small note that overall we noticed no concerns over the correctness of our justifications or reasoning and the steps we took to derive our contributions. We think this is one of the strongest points in favor of our work and well differentiates us from prior work, which lacked justification for a number of aspects and did not go far enough to ensure or verify quality in the data and evaluation procedures. We believe the principled approach taken during the creation of the dataset and metrics proposed in this work makes them highly valuable contributions to the research community and provides a foundation on which to move progress forward in this area.

One of the major points raised by the reviewers was that the exact process used to create the data, specifically the canonicalization and correction process that improves the original table authors’ annotations, was not described in enough detail to ensure reproducibility. We agree that this is important because the canonicalization process is one of the major contributions of the paper, and full details were omitted originally only due to page constraints. In response, we have included an appendix where we describe the canonicalization procedure step-by-step. In addition, we plan to release the code for the entire dataset creation process along with the camera-ready version of the paper, which we agree ensures reproducibility and strengthens the contribution made by the paper.

Another of the major points raised appears to be that while the paper provides enough background and theoretical justification to demonstrate correctness, the chain of reasoning required to put the results into context may be too long. In other words, more could be done to make it easier for readers to interpret the results and assess the value of the contributions. For example, the baseline model we describe appears to be quite strong according to our own proposed metrics, but more context would be helpful to draw conclusions, including: more evidence that the problem is hard and more evidence that the metrics are meaningful. This would be helpful to demonstrate more clearly that DETR’s performance on the tasks and the conclusions we draw from its performance are meaningful.

In response, we have used the additional allowed page to include additional experiments in the current revision that show the metrics are well-behaved and well-distinguish good models from bad models. We have also provided more citations to papers that outline the inherent difficulty of the table extraction problem, to better motivate the importance of our work. In the camera-ready version of the paper, we will include three additional experiments: 1. Training and evaluating Faster R-CNN on the same data, which will show more clearly that the dataset is not trivial to learn; 2. Training DETR on 1/10th of the data and evaluating it on the full test set, which will help to answer how important the size of the dataset is; 3. Training DETR on the unprocessed (non-corrected, non-canonicalized) annotations, which will help to show how much the canonicalization and correction process contributes to reducing noise in the training and evaluation. We believe these experiments address the feedback about assessing the relative impacts of our different contributions.

In the current revision, in addition to the improvements described above, we have also updated the datasheet and added a Future Work section in response to reviewer feedback.

More details for these improvements and all the additional improvements we plan to make within the camera-ready version of the paper are addressed in our individual responses to the reviewers.

---

### Decision · Program_Chairs · 2021-07-26

**Decision:**

Reject

**Comment:**

In this paper the authors offer an expanded dataset and evaluations for extracting tables from unstructured documents.  They apply this approach to PubMed and show that it works well, providing a large scale new dataset for the task.  Consistent feedback across reviewers was that while the results seemed impressive, the clarity of the approach and the significance could be significantly improved.  I'd encourage the authors to continue to polish the paper for clarity so that it can reach a broader audience.  I'd encourage the authors to revise and resubmit.